

# Increased absorption in autonomous sensory meridian response

Agnieszka B. Janik McErlean and  Eleanor J. Osborne-Ford

School of Science, Bath Spa University, Bath, United Kingdom

## ABSTRACT

**Background**.  Autonomous sensory meridian response (ASMR) is a cross-sensory phenomenon characterised by a static-like sensation which typically originates on the scalp and spreads throughout the body leading to a state of deep relaxation. It can be triggered by visual and auditory stimuli in real life, incidentally by various media and via intentionally created ASMR videos. Previously ASMR has been linked to a specific personality profile and this study aimed to further elucidate individual differences associated with this phenomenon.

**Methods**. To this effect ASMR-Experiencers and age and gender matched controls were compared on measures of flow, absorption and mindfulness.

**Results**. This revealed that ASMR was associated with elevated absorption but no group differences were found with respect to the other constructs, suggesting that the ability to get deeply immersed with the current experience accompanied by loss of reflective awareness may be an important factor contributing to the experience of ASMR.

## INTRODUCTION

Autonomous sensory meridian response (ASMR) is a multisensory phenomenon where auditory and visual stimuli such as whisper or personal attention trigger a pleasant, static-like tingling sensation which typically originates from the head and disperses throughout the body resulting in a relaxed state (*Barratt & Davis, 2015*). Although ASMR can be experienced in daily life, in recent years many ASMR-inducing YouTube channels have been created attracting large audiences who watch such videos to experience the sensation, relax, and fall asleep and in some cases to alleviate anxiety and stress (*Barratt & Davis, 2015*; *Janik McErlean & Banissy, 2017*; *Poerio et al., 2018*). Although the prevalence of ASMR is not known, the popularity of ASMR media suggests that the phenomenon might be widespread.

One line of evidence suggests that there are personality differences between people who experience ASMR and those who do not. Specifically, trait ASMR has been linked to (i) increased openness to experience (*Janik McErlean & Banissy, 2017*; *Fredborg, Clark & Smith, 2017*) which taps into one's interest in novel experiences and propensity to fantasise (*John, Naumann & Soto, 2008*), (ii) heightened fantasising (*Janik McErlean & Banissy, 2017*), which reflects an ability to immerse oneself in a fictional reality (*Davis, 1983*), and (iii) elevated mindfulness (*Fredborg, Clark & Smith, 2018*), which entails concentrating

Corresponding author
Agnieszka B. Janik McErlean,
a.janikmcerlean@bathspa.ac.uk

on the present moment (*Brown & Ryan, 2003*). These findings are interesting taking into account that for many individuals ASMR is triggered when focusing on the external triggers which resembles mindfulness practice (*Fredborg, Clark & Smith, 2018*) and that ASMR videos, which often entail role plays, require imaginatively transposing oneself into the virtual reality (*Janik McErlean & Banissy, 2017*).

ASMR has also been previously likened to a state of flow (*Barratt & Davis, 2015*), which can be measured both as a global construct and in terms of its underlying components including warped passing of time (*Ross & Keiser, 2014*), complete absorption (*Jackson et al., 2001*), and intense concentration when fully engaged in optimally challenging and intrinsically rewarding tasks (*Csikszentmihalyi & Csikszentmihalyi, 1992*). Flow has also been conceptualised as a trait, whereby individuals with the so called autotelic personality (who are intrinsically motivated) are more susceptible to experiencing flow (*Csikszentmihalyi, 2000*). To date several measures have been developed which allow for measuring flow both as a state and a trait (e.g., *Jackson & Eklund, 2002*; *Jackson & Eklund, 2004*). *Barratt & Davis (2015)* suggested that ASMR is a 'flow-like' phenomenon achieved by watching others in a similar state. Interestingly, some of the most popular ASMR triggers, such as watching someone making expert hand movements, are typical examples of being in a state of flow (*Janik McErlean & Banissy, 2017*). Moreover, individuals with greater susceptibility to flow have been found to report a greater number of ASMR triggers highlighting a link between the two phenomena (*Barratt & Davis, 2015*). Furthermore, a positive association between flow and a newly developed ASMR measure has been recently reported (*Roberts, Beath & Boag, 2019*). However, whether ASMR is associated with increased levels of flow both in terms of intensity and prevalence compared to the general population is currently not known. In addition, both *Barratt & Davis (2015)* and *Roberts, Beath & Boag (2019)* used a modified version of the Flow State Scale (*Jackson & Marsh, 1996*) which only taps into the passive aspects of flow. While this measure is more likely to be relevant in the ASMR context which does not entail active engagement in activities, it does not capture the other core components of the flow experience such as the balance between challenge and skill. As such no study to date examined whether ASMR is associated with increased levels of flow using the global measure of this construct.

The reported link between ASMR and flow is also interesting in the context of increased mindfulness among ASMR-experiencers (*Fredborg, Clark & Smith, 2018*). Mindfulness, similarly to flow, can be conceptualised as a trait and as a state. In *Fredborg, Clark & Smith (2018)* study ASMR-experiencers were found to score higher than controls in terms of trait mindfulness based on their scores on the Mindful Attention Awareness Scale which taps into one's general disposition to pay attention to and be aware of the present moment (*Brown & Ryan, 2003*). They also scored higher on the Curiosity subscale of the state mindfulness measure called the Toronto Mindfulness Scale which suggests increased interest in one's own inner experiences among the ASMR group (*Lau et al., 2006*). Although mindfulness and flow are similar in terms of concentrating on the present experience, they differ dramatically in terms of the role of self-awareness which is enhanced in mindfulness and diminished in flow where loss of self-consciousness is the central feature (*Bishop, 2002*; *Bishop et al., 2004*; *Nakamura & Csikszentmihalyi, 2009*). Moreover,

although existing research provides support for the positive association between the global measure of flow comprising all underlying dimensions and mindfulness (*Thienot et al., 2014*), recent findings suggest that this is driven only by the control facet of flow which reflects a sense of agency and mastery over a task, whereas absorption which is another core underlying dimension of flow is in fact negatively related to mindfulness (*Sheldon, Prentice & Halusic, 2015*).

Absorption as a key facet of flow is a trance-like state of consciousness characterised by an ability to fully focus one's attention on a particular object or situation and to become perceptually engrossed with the current experience (*Tellegen, 1981*; *Tellegen & Atkinson, 1974*). It is also a stable personality trait typically measured with the Tellegen Absorption Scale (*Tellegen & Atkinson, 1974*). Absorption has also been linked to hypnotisability, imagery, day-dreaming, and openness to experience (*Weilbel, Wissmath & Mast, 2010*; *Glisky et al., 1991*). Recently, an association between absorption propensity and ASMR has also been reported (*Roberts, Beath & Boag, 2019*). Taking into account existing findings and the phenomenological similarities between absorption and the immersive nature of ASMR, one may expect increased levels of absorption among ASMR-experiencers compared to controls. However, considering previous reports of increased mindfulness in ASMR (*Fredborg, Clark & Smith, 2018*) and the conceptual incompatibility between mindfulness and absorption (*Sheldon, Prentice & Halusic, 2015*) it is essential to examine these constructs together in the context of ASMR.

The purpose of this study was to investigate whether there are differences between ASMR-experiencers and controls in terms of flow, absorption and mindfulness. This study also aimed to elucidate the relationship between these constructs and ASMR characteristics such as intensity, pleasantness, and ASMR videos viewing habits.

## MATERIALS & METHODS

### Participants

Participants were recruited via websites dedicated to ASMR and among Psychology students, who were offered credits for their participation. Participants were asked to indicate whether they would classify themselves as controls or ASMR-experiencers based on the description of the phenomenon (as per *Janik McErlean & Banissy, 2017*). To further verify that participants in the ASMR group were genuine ASMR-experiencers they were asked to provide a series of answers related to their ASMR experience (see Questionnaire). E.g., they were asked to indicate the intensity of their ASMR when engaging with various stimuli. If a participant indicated they did not experience ASMR in response to any of the popular triggers they were excluded from the analysis. There were only two such individuals, who also happened to be outliers in terms of their age. They were excluded from the analysis. A priori power analysis revealed that suggested sample size to conduct MANOVA was 232 participants in order to obtain statistical power at 0.90 level (Effect size = 0.06, $\alpha$ = 0.05). Through opportunity sampling, a total of 316 participants completed the survey (59% = ASMR-experiencers). In order to reduce the imbalance between the group sizes and to match the two groups in terms of age and gender 124 ASMR-experiencers (92

female, 30 male, 2 other, age $M = 21.79$, SD $= 3.36$) and 124 controls (92 female, 30 male, 2 other, age $M = 21.40$, SD $= 3.13$) were selected from the total sample prior to data analysis resulting in the final sample of 248 individuals which was used to compare the groups on absorption, flow and mindfulness. These participants were first matched on gender and then within gender on age $+/-$ 2 years. Majority was matched within $+/-$ 1 year and four participants who identified as non-binary in terms of gender were matched within $+/-$ 4 years.

In addition, data contributed by those ASMR-experiencers who stated they watch ASMR videos ($N = 149$, 121 female, 26 male, 2 other, age $M = 26.22$, SD $= 8.12$) was used in the correlational analysis which aimed to examine the relationship between absorption, flow and mindfulness and ASMR characteristics including pleasantness, intensity, number of videos watched in a single session and frequency of watching ASMR videos.

## Measures

The study employed an anonymous survey which was administered via Bristol Online Survey and was approved by the Ethics Reviewers at Bath Spa University. All participants were asked to provide an electronic consent prior to taking this survey.

Tellegen Absorption Scale (TAS) employed in this study is a widely used measure of absorption (*Tellegen & Atkinson, 1974*). It consists of 34 questions and the participants are asked to indicate the degree to which they agree with each statement such as 'While watching a movie, TV show or a play, I may become so involved that I may forget about myself and my surroundings and experience the story as if it were real and as if I were taking part in it' on a scale from 1 (disagree strongly) to 5 (agree strongly). A total score is calculated by summing the scores for each question. Internal consistency of TAS in this study was $\alpha = .93$.

Mindful Attention and Awareness Scale (MAAS) is an established measure of mindfulness (*Brown & Ryan, 2003*). It consists of 15 items such as 'I could be experiencing some emotion and not be conscious of it until sometime later'. Participants are asked to indicate how frequently they have such experiences on a scale from 1 (almost always) to 6 (almost never). The final MAAS score is calculated by averaging all individual answers. The internal consistency of MAAS in this study was $\alpha = .79$.

Flow Questionnaire: Firstly, participants were presented with the description of flow (as per *Csikszentmihalyi, 1988*; *Csikszentmihalyi, 1990*) based on which they indicated whether they have ever had similar experiences which allows for classifying individuals into those capable of experiencing flow and those who are not. In order to avoid false positive answers those who reported to experience flow were also asked to state what activities they were engaged in when having such experiences. Subsequently, they were asked to complete the Flow Experiences Scale (FES; *Schwartz & Waterman, 2006*) which consists of eight statements corresponding to those originally outlined by *Csikszentmihalyi (1988)* and *Csikszentmihalyi (1990)* which inquire about different aspects of the experience of flow including e.g., 'I lose track of time' to which participants provide a rating on a scale from 1—not at all characteristic of me to 7—very characteristic of me. A total score reflecting the

extent to which flow is experienced is calculated by summing the scores for each statement. Question 2 is reverse scored. The internal consistency for FES in this study was $\alpha = .74$.

ASMR Questionnaire: In addition, ASMR-experiencers completed the self-designed ASMR questionnaire which was based on items previously used in ASMR literature (*Barratt & Davis, 2015*; *Janik McErlean & Banissy, 2017*; *Fredborg, Clark & Smith, 2018*). Specifically, participants were asked: (1) whether they watch ASMR videos (Yes/No), (2) why they watch ASMR videos (open ended), (3) how often they watch ASMR videos: never, less than once a month, 2–3 times a month, 2–3 times a week, daily, (4) whether they require specific conditions to achieve ASMR (open ended), (5) to indicate the intensity of common triggers such as crisp sounds, personal attention etc. using a scale from 0 (no tingles) to 6 (the most intense tingles)—ASMR intensity score was calculated by averaging the scores across all listed triggers as per *Fredborg, Clark & Smith (2018)*, (6) whether the intensity of these triggers varies from session to session (Yes/No); (7) to indicate how pleasurable ASMR is from 1—Quite Uncomfortable to 5—Quite pleasurable. Open ended questions were coded prior to the analysis.

## RESULTS

### Data inspection and assumptions testing

The data was normally distributed and no outliers were identified as all values fell within $+/-2.5$ SD from the mean. However, as only 106 out of 248 (43%) participants reported to experience flow it was not possible to conduct MANOVA due to the insufficient sample size. Instead three separate independent samples t-tests were conducted to compare the groups on the three personality traits.

### Comparison of ASMR-experiencers and controls on personality traits

Three independent-samples $t$-tests revealed: (i) a non-significant group difference $t(104) = -0.990$, $p = .324$, $d = 0.19$, 95% CI [$-4.807$; 1.606] between ASMR-experiencers ($M = 36.49$, SD $= 8.16$) and controls ($M = 38.09$, SD $= 8.19$) in terms of flow, (ii) a non-significant group difference on mindfulness $t(246) = -0.715$, $p = .475$, $d = 0.09$, 95% CI [$-0.273$; 0.127] between ASMR-experiencers ($M = 3.20$, SD $= 0.80$) and controls ($M = 3.28$, SD $= 0.79$), (iii) a statistically significant group difference on absorption $t(246) = 4.995$, $p < .001$, $d = 0.63$, 95% CI [8.979; 20.252] with ASMR-experiencers ($M = 108.74$, SD $= 23.80$) scoring higher than controls ($M = 94.22$, SD $= 21.95$) (see Table 1 for Descriptive Statistics). Furthermore, chi-square analysis revealed that the proportion of individuals who reported to experience flow was significantly greater [$\chi^{2(1,N=248)} = 6.590$, $p = .010$] among ASMR-Experiencers (51%) compared to controls (34%).

An additional analysis was conducted using the whole data set rather than just the subsample of ASMR-experiencers and control participants who were matched for age and gender. This analysis revealed qualitatively similar results to the ones reported above (see Supplemental Results).
**Table 1  Means and standard deviations for ASMR-experiencers and controls on FES, MAAS and TAS.**

| Variable | ASMR | | Control | |
|---|---|---|---|---|
| | **Mean** | **SD** | **Mean** | **SD** |
| FES | 36.49 | 8.16 | 38.09 | 8.19 |
| MAAS | 3.20 | 0.80 | 3.28 | 0.79 |
| TAS | 108.74 | 23.80 | 94.22 | 21.95 |

**Notes.**
FES, Flow Experiences Scale; MAAS, Mindful Attention and Awareness Scale; TAS, Tellegen Absorption Scale.
N (MAAS, TAS): ASMR = 124, Control = 124; N (FES): ASMR = 63, Control = 43.

**Table 2  Correlations between ASMR intensity, number of videos watched in a single session, frequency of watching ASMR videos, pleasantness of ASMR and the scores on TAS, MAAS and FES.**

| Variable | Intensity | Number | Frequency | Pleasantness | TAS | MAAS | FES |
|---|---|---|---|---|---|---|---|
| Intensity | – | −.005 | .354*** | .389*** | .180* | .033 | .207 |
| Number | −.005 | – | .111 | −.030 | −.061 | −.040 | −.232 |
| Frequency | .354*** | .111 | – | .233** | −.009 | −.072 | .017 |
| Pleasantness | .389*** | −.030 | .233** | – | .156 | −.042 | .168 |
| TAS | .180* | −.061 | −.009 | .156 | – | −.027 | .156 |
| MAAS | .033 | −.040 | −.072 | −.042 | −.027 | – | .173 |
| FES | .207 | −.232 | .017 | .168 | .156 | .173 | – |

**Notes.**
N (Intensity, Frequency, Pleasantness, TAS MAAS) = 149. N (Number) = 143. N (FES) = 72.
TAS, Tellegen Absorption Scale; MAAS, Mindful Attention and Awareness Scale; FES, Flow Experiences Scale.
*$p < .05$.
**$p < .01$.
***$p < .001$.

## Correlations between ASMR features and personality traits

Pearson's correlations employed to examine the relationship between absorption, flow and mindfulness and ASMR characteristics revealed a positive relationship between: (i) absorption and the intensity of ASMR ($r(149) = .180, p = .028$), (ii) intensity and frequency of watching ASMR videos ($r(149) = .354, p < .001$), (iii) intensity and pleasantness ($r(149) = .389, p < .001$) and (iv) pleasantness and frequency of watching ASMR videos ($r(149) = .233, p = .004$). No other correlations were statistically significant (Table 2; Fig. 1).

## Summary of responses to ASMR questionnaire

The main motivation behind watching ASMR videos was to relax (71%), followed by to fall asleep (60.1%), to experience ASMR (54.1%), to improve mood, especially in relation to anxiety and depression (12.8%), and to help concentrate on work or a task (6.8%). On average, participants watched 3.3 videos per session. 42.3% reported to watch ASMR videos daily, followed by 2–3 times a week (30.9%), 2–3 times a month (19.5%), and less than once a month (7.4%). 49% of participants stated that they needed specific conditions to experience ASMR. This included: a quiet room (71.23%), dim lighting (30.14%), to be alone or have no distractions (26.03%), to be wearing headphones (20.55%), to be in bed or lying down (13.7%), to have the room at a specific temperature (12.33%) to be in a
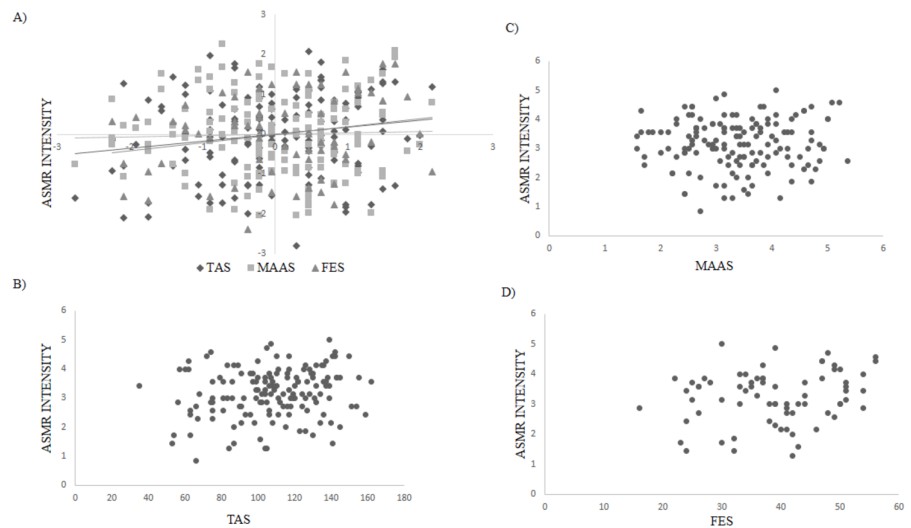

**Figure 1** (A) Relationship between ASMR Intensity and absorption (TAS), mindfulness (MAAS), and flow (FES) based on *z* scores; (B) relationship between ASMR Intensity and TAS based on original scores; (C) relationship between ASMR Intensity and MAAS based on original scores; (D) relationship between ASMR Intensity and FES based on original scores.

real life setting (8.22%), comfort (8.22%), which is largely consistent with the findings of *Barratt, Spence & Davis (2017)*. In terms of the intensity on average whispering ($M = 4.03$) was rated to produce the most intense ASMR, followed by personal attention ($M = 3.65$), crisp sounds ($M = 3.57$), paying attention to detail, concentrating on something, slowly performing mundane actions or explaining something ($M = 3.43$), role-play ($M = 3.15$), hair brushing ($M = 3.04$) and lastly people eating ($M = 1.34$). 94% reported that the intensity of ASMR varies from session to session. The majority of participants (60.4%) gave ASMR the highest rating in terms of how pleasurable it is, followed by 35.6% rating ASMR as mildly pleasant, 2.7% as neutral, one participant reported ASMR to be quite uncomfortable (0.7%) and another one as mildly uncomfortable (0.7%).

## DISCUSSION

The purpose of this study was to further elucidate whether ASMR is associated with wider differences in terms of personality traits. Current results demonstrate that ASMR-experiencers show elevated absorption compared to age and gender matched controls, but no group differences were found in terms of mindfulness or flow. This suggests that ASMR-experiencers display increased readiness for experiential involvement and heightened ability to become fully engaged with the current experience (*Tellegen, 1981*; *Jamieson, 2005*). This is interesting taking into account that being immersed in the virtual reality appears to be a key component of technologically-mediated ASMR and that increased absorption has been previously linked to a more immersive virtual reality experience (*Baños et al., 1999*). Moreover, increased absorption has been associated with elevated openness to experience (*Weilbel, Wissmath & Mast, 2010*; *Glisky et al., 1991*)

which is also heightened among ASMR-experiencers (*Janik McErlean & Banissy, 2017*; *Fredborg, Clark & Smith, 2017*). ASMR has also been linked to another trait relevant to technologically-mediated ASMR i.e., increased fantasizing (*Janik McErlean & Banissy, 2017*) which reflects heightened propensity to become imaginatively involved in a fictional reality. As such, the combination of increased absorption, openness to experience and fantasizing may contribute to the individual likelihood of experiencing ASMR. Moreover, when considering those individuals who consume ASMR media to alleviate stress and anxiety it is possible that for them becoming fully absorbed in ASMR videos may act as a form of distraction from psychological distress which is similar to the well documented effect of virtual reality gaming and other immersive methods as effective pain reduction strategies (*Dahlquist et al., 2007*; *Jameson, Trevena & Swain, 2011*).

The current study also found that ASMR-experiencers did not differ from controls in terms of mindfulness, which is consistent with our results of increased absorption among ASMR-experiencers and the reports of conceptual incompatibility between the two constructs (*Brown & Ryan, 2003*; *Sheldon, Prentice & Halusic, 2015*). Moreover, the correlational analysis employed in this study revealed no association between mindfulness and absorption further suggesting little overlap between these psychological variables. It therefore appears that it is the total immersion in the experience accompanied by a loss of self-awareness, which are core features of absorption, rather than sustained consciousness of the current moment characteristic of mindfulness that are features relevant to the ASMR experience. However, it is of note that these results are inconsistent with previous findings of increased mindfulness in ASMR (*Fredborg, Clark & Smith, 2018*). As both studies employed the same scale and participant recruitment method it is not clear why different results were obtained. Although the sample size in this study was not as large as in the other one, it was sufficiently large as determined by a priori power analysis. However, it appears that the control participants sampled by *Fredborg, Clark & Smith (2018)* scored particularly low on MAAS ($M = 3.02$, SD $= 1.17$) compared to the control participants in the present study ($M = 3.28$, SD $= 0.79$), which might explain the difference in results between the two studies. Moreover, both the current and *Fredborg, Clark & Smith (2018)*'s results pertaining to the control sample are much lower than originally reported by the MAAS authors who found average MAAS scores to be $M = 3.85$ (SD $= 0.68$) among undergraduate students and $M = 3.97$ (SD $= 0.64$) in a community sample (*Brown & Ryan, 2003*). It is not clear what drives this discrepancy in findings between the three studies. As such, the potential link between ASMR and mindfulness should be further explored in future research.

Furthermore, current results show similar levels of flow between ASMR-experiencers and controls. This is consistent with existing literature demonstrating an association between the global flow construct and mindfulness (*Thienot et al., 2014*) and current findings of no group differences on either of these traits. This is also in line with the original conceptualisation of the construct which emphasises that active engagement in activities is necessary for the experience of flow (*Csikszentmihalyi & Csikszentmihalyi, 1992*) and research showing that flow has been most commonly related to activities that are effortful and based on challenge, skill and intrinsic motivation (*Mauri et al., 2011*) whilst the experience of ASMR is typically elicited through passively watching videos

or by observing people in real life who engage in ASMR-inducing activities. Therefore, ASMR appears to only reflect the passive aspects of flow, which are more consistent with the construct of absorption (*Jackson et al., 2001*). This is also in line with *Barratt & Davis (2015)* and *Roberts, Beath & Boag (2019)* studies who measured flow in terms of its passive component which was found to be positively associated with the ASMR experience. Moreover, these results also fit with previous reports of elevated fantasising and imaginative involvement in ASMR (*Janik McErlean & Banissy, 2017*) and the fact that fantasy engagement and imagination are key characteristics of absorption but not flow (*Nakamura & Csikszentmihalyi, 2014*). However, it is of note that substantially more ASMR-experiencers reported to experience flow (51%) compared to controls (34%) suggesting that while the extent to which both groups experience flow is similar, flow appears to be more prevalent among ASMR-experiencers. Interestingly, the majority of ASMR sample reported to experience flow when engaged in sport, music, art or when concentrating on a work-related task which are typical examples of flow-inducing activities (*Csikszentmihalyi & Rathunde, 1993*) and only five individuals reported to experience flow when consuming ASMR media. This suggests that ASMR group appears to experience flow more readily than controls but rarely in an ASMR context. However, it is of note that although previous research is quite inconsistent when it comes to the prevalence of flow, the percentage of individuals identifying with the experience in this study (both ASMR-experiencers and controls) is comparably low. For instance, *Moneta (2012)* reports that about two-thirds of the general population are capable of experiencing flow while some of the earlier studies suggest the experience to be universal with prevalence rates of 97% among university students (*Massimini, Csikszentmihalyi & Delle Fave, 1988*). Although, *Han (1988)* reports that only 33% of elderly Korean immigrants in America identified with the experience suggesting potential effects of age and culture.

This study also found that absorption was positively correlated with the intensity of ASMR suggesting that this trait is linked to the extent to which ASMR is experienced. However, no association was found between flow and mindfulness and ASMR features, further demonstrating that these traits are not related to ASMR. In addition to this, our results suggest that those who experience ASMR more intensely find it more pleasurable and also engage with ASMR media more frequently. However, it is important to note that due to the correlational nature of this research it is not possible to conclude that the trait of absorption predisposes people to experience ASMR or that the ability to experience ASMR increases the propensity for absorption. It is also possible that both absorption and ASMR are underlined by a third variable such as openness to experience.

Furthermore, although one may suspect that the self-selection bias may have contributed to the current results of increased absorption among ASMR-experiencers compared to controls as 75% of the ASMR sample whose data was used for comparison purposes and overall 80% of the entire ASMR sample in this study (not just the ones matched for age and gender) reported to watch ASMR-videos. However, we have found that ASMR-experiencers who watch ASMR videos ($M = 108.31$, $M = 27.27$) and ASMR-experiencers who do not watch such videos ($M = 108.04$, SD = 25.679) reported exactly the same level absorption

suggesting that our results pertain also to those ASMR-experiencers who do not engage with ASMR media.

In addition, consistently with previous research this study has found that the key reasons for watching ASMR videos were to relax, fall asleep, to experience ASMR, and to help with stress or anxiety (*Barratt & Davis, 2015*; *Janik McErlean & Banissy, 2017*). In addition, 6.7% of participants reported to play ASMR videos in the background while studying or working. Moreover, 49% of participants stated that they needed specific conditions, such as a quite space with few distractions, in order to experience ASMR which is consistent with the previous studies (*Barratt & Davis, 2015*).

## CONCLUSIONS

In summary, our findings contribute to the existing literature documenting individual differences associated with ASMR by showing that absorption proneness may be an important factor contributing to the experience of ASMR. This study also shows that ASMR is not linked to the constructs of flow and mindfulness.

### Funding
The authors received no funding for this work.

### Competing Interests
The authors declare there are no competing interests.

### Author Contributions
- Agnieszka B. Janik McErlean conceived and designed the experiments, analyzed the data, prepared figures and/or tables, authored or reviewed drafts of the paper, and approved the final draft.
- Eleanor J. Osborne-Ford performed the experiments, analyzed the data, prepared figures and/or tables, authored or reviewed drafts of the paper, and approved the final draft.

### Human Ethics
The following information was supplied relating to ethical approvals (i.e., approving body and any reference numbers):

This study was approved by the Ethics Reviewers at Bath Spa University.

### Data Availability
Output containing raw data and computed/coded final scores is available in the Supplemental Files.

### Supplemental Information
Supplemental information for this article can be found online at http://dx.doi.org/10.7717/peerj.8588#supplemental-information.

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
