# Peer review of "Increased absorption in autonomous sensory meridian response"

_PeerJ, doi:10.7717/peerj.8588_

## Round 0.1 · original submission · Minor Revisions

Thank you for your submission. This has been well received by all three reviewers, who have made a number of suggestions to enhance your paper. These should all be addressed in your next version. In particular you should work on your introduction section to add further detail on mindfulness and flow.

·

Basic reporting

The standard of this manuscript is high, and I have no concerns with the quality of the writing, of the visual elements, or the overall structure.

Experimental design

I have no concerns with the experimental design. However I have a (completely unsupported) suspicion that everyone is potentially an ASMR-experiencer, so in that sense there is no control group. However I like the use of a questionnaire to classify people.

Validity of the findings

No problem here (but see next section for comments).

Additional comments

1. Line 117. How were the included participants selected? Was this random? Have the authors tried the analysis without dropping some participants, and if so did it change the findings?
2. Line 142. The flow questionnaire in use here is different to the one used by Barratt & Davis (2015). Why the difference? Barratt & Davis did not spend long looking for appropriate scales for that paper, so it would be useful to know if the FES is a better instrument.
3. Line 185. The relationship between ASMR and the personality measures is presented here as a table (Table 2). It would aid interpretation if these were also presented as a scatter plot, since that would reveal possible non-linear structure in the data.
4. Line 203. Although the authors should not feel the need to cite it, Barratt, Spence & Davis (2017) PeerJ 5:e3846, found that the sorts of video features that trigger ASMR accord well with the list provided here. The surprise (to the authors) in that work was that background music is highly distracting for ASMR.
5. Line 230. 'AMR' should be 'ASMR'.
6. Line 243. The inconsistency with the Fredborg work is very interesting. I wonder if ASMR is not a form of mindfulness at all, as there is a clear, external stimulus to focus on (my understanding of mindfulness is that the focus is internal). I agree with the authors that more work is needed, as the two constructs are often linked in the popular media.

·

Basic reporting

The paper reports the results of a single study investigating the role of absorption, flow, and mindfulness in Autonomous Sensory Meridian Response (ASMR). A questionnaire was administered to participants who experience ASMR, as well as those that do not. The primary measures were absorption, mindfulness, and flow. The results showed that the groups differed on the measure of absorption, but not flow or mindfulness. The main implication is that it is the passive ability of absorption that contributes to the ASMR experience, compared to more ‘active’ elements of flow or mindfulness. The paper contributes to the growing literature on ASMR and possible causes of the phenomenon.

The reporting of the study is good, the article is clearly structured and well written overall. There is sufficient background information about ASMR, and the logical structure of the introduction is good: the purpose of the research is clear and well-justified, and the paper presents the results of a clear ‘self-contained’ which sets out a clear research question and answers it. Raw data is available and clearly labelled.

Although the authors have done a reasonable job of explaining flow, mindfulness, and absorption, I didn’t feel confident understanding the differences between the constructs after reading the article. I think more clarity throughout on both (a) the differences between the constructs, and (b) the measurement of these constructs, would improve the piece and really help the explanation of the work.

For example, when flow is introduced in the introduction, the current explanation leaves several things unclear. Is “flow” a state or a trait? does it describe a singular experience or a trait like individual difference? it is unclear what scoring “high on flow” (line 74) really means – people are more susceptible to flow? Or they experience it more often? What is the distinction between a regular measure of flow and a “global measure of flow” (line 83)? Considering the conclusions of the research concern the differences between constructs which are at face value very similar, it is important to clearly describe and explain the differences between the constructs and their measurements in the introduction.

This issue particularly applies to the construct of mindfulness, which has been criticized for being particularly nebulous and multifaceted, and sometimes even meaning completely contradictory things! A useful paper for the authors to use to help define mindfulness in their introduction is:

Van Dam, van Vugt, Vago, Schmalz… et al. (2018) Mind the Hype: A critical Evaluation and Prescriptive Agenda for Research on Mindfulness and Meditation. Perspectives on Psychological Science, 13, 36-61. https://doi.org/10.1177/1745691617709589


A clear definition of mindfulness would also help with justifying the study. I liked the explanation at the end of the introduction indicating the incompatibility of previous results, and explaining how investigating the role of absorption would elucidate this. I think with clearer definitions of flow and mindfulness this would be improved further.
The definition of absorption is clear, and it was helpful to me when you described it as a “stable personality trait”. The structure of this section could be improved in to more clearly indicate that absorption is a facet of flow and clarify the relationship between flow and absorption.

This extra clarity provided in the introduction regarding the concepts of flow, absorption and mindfulness and their relationship can also be carried over into the discussion when explaining the findings (e.g. lines 242-243; 251-259). On reading the article, it was only when I got to this section (e.g. lines 256-258) that I began to understand the relationships between the variables, and I think moving this clarity into the introduction would help.

Regarding other reporting, There are no statistical errors in the reporting of the inferential statistics, as determined by http://statcheck.io . The descriptive statistics in Table 1, however, do not correspond to those reported in the results section. For example, SD for absorption in ASMR participants is 23.81 in the text, and 23.80 in the table. Mindfulness mean/SD for ASMR participants is 3.21/.81 in the text, 3.20/.80 in the table. For flow, mean for ASMR is 36.49 in text and 36.5 in the table; and 38.16 in the text and 38.09 in the table for controls. These are just the errors I noticed, Please can the authors double check all numbers in all tables to ensure consistency with the text, and the statistics calculated from the uploaded raw data.

Regarding reporting style of statistics, the authors should use a consistent reporting style. In some places, p values are reported as <.05 (e.g. line 188), in others they are reported in full (e.g. line 182). Likewise, in some places they are reported to 2 decimal places (line 177) and in others 3 (line 179). I would suggest the authors follow APA style, which dictates that all p values are reported in full to 3 decimal places, the only exception being very small p values reported as <.001. Please can the authors go through the manuscript and change the reporting for all these statistics.

Another (pedantic) point regarding reporting statistics: APA style recommends that leading zeros be used when reporting statistics that can feasibly be larger than 1, but not for statistics that cannot (see https://blog.apastyle.org/apastyle/2010/07/a-post-about-nothing.html). Therefore, the SD’s reported in table 1 (and in the results section, e.g. line 178), and the Cohen’s D values reported in the results section (e.g. line 175) should have leading zeroes (e.g. 0.81 rather than .81), as the values can potentially exceed 1.

Finally a few bits of language I found awkward/misleading/debatable:

Abstract: “intentionally created ASMR Youtube videos” – Whilst most ASMR videos are created intentionally, there is also a large number that trigger ASMR incidentally (e.g. Bob Ross videos) so I think this is misleading.

Intro, lines 62-65: Whilst I agree that there is an element of focus in experiencing ASMR, the description that it is necessary to focus on certain triggers to experience it doesn’t sit well with the “autonomous” description of the sensation, and I’m not sure whether it is maybe a bit misleading to say that one needs to focus on the triggers in order to experience it. E.g. experiencing it in everyday life – one does not need to prepare to focus on a trigger in order to subsequently experience ASMR, - it may just happen autonomously in response to triggers – although I do agree that subsequent focusing may prolong and intensify the sensation. So perhaps this could be reworded to just clarify that one does not have to choose to focus on the triggers in order to experience it at all.

Line 84 – I would like a comma after the (Thienot et al., 2014) reference
Line 95 – missing “the” before “immersive nature of ASMR”

Experimental design

The study reports original primary research within the scope of the journal. The research question is well defined, relevant, and meaningful, and the paper makes an excellent contribution to the growing literature on ASMR.

Enough information is included to replicate the study, and the methods are described with good detail. My only comment echoes the one in section 1 regarding clarifying the concepts of flow and mindfulness and their measurement, which has impacts for understanding the purpose of the study, the findings, and the methodology with regards to the constructs being measured. The following paper gives advice about reporting measurement and the relationship between constructs which the authors might find useful, e.g:

https://psyarxiv.com/hs7wm/

Validity of the findings

The statistical tests performed are appropriate, and replicable using the dataset provided.

One interesting question concerns the incongruence of the present findings and Fredborg et als (2017). I was intrigued by this, especially as both studies used the same measure of mindfulness. There appears to be no obvious difference between the sampling method, and both studies were adequately powered, as discussed by the authors.

However, the descriptive statistics might give us a clue. The means for mindfulness in ASMR participants in both studies are actually very similar: 3.34 (SD= 0.75) in the Fredborg data, and 3.21 (SD = 0.81) in the current data. It is the difference in the control participants that is more obvious. In the Fredborg study, the control participants had a mindfulness score of 3.02 (SD = 1.17) - but in the present study, the control participants had a score of 3.28 (SD = 0.79). Investigating the raw data, I created histograms of the four distributions using the original data. (e.g. ASMR and control participants, for both studies, attached as a supporting document).

It seems that in the Fredborg study, the control group sampled a particularly high number of people with low MAAS scores (indicated by the positively skewed distribution), whereas in the present study, the control participant data appears to be slightly negatively skewed (indicating more controls reporting higher mindfulness scores, and very few controls reporting low mindfulness scores). The distributions for ASMR participants in both studies are normally-distributed.

It seems to me that the difference in the means and distributions of the control participant scores could explain the discrepant findings between the present research and Fredborgs, and (if the authors agree!) this could be presented as a potential explanation.
I agree with the recommendation to encourage further research on this question.


A second point I would like to see slightly more discussion of in the discussion is Flow, and its measurement. It was interesting to see that only 43% of participants reported Flow. Is this normal? How does this compare to other studies that have used this measure of flow? Have they also found similarly low percentage of people reporting it? If the amount is not similar to those reported in previous research, are there any reasons for this? A brief discussion on this would be helpful for other researchers reading the paper who may be interested in researching the relationship between ASMR and flow.

Additional comments

Overall the study makes an excellent contribution to the growing literature on ASMR, and the finding that absorption is a key element of the ASMR experience. In order of priority, the main suggestions to improve the manuscript would be:

1. To employ a clearer definition of the construct of mindfuless in the introduction, and clearer discussion about how this construct (in the particular way the authors are conceptualizing it and measuring it with the MAAS) is related to flow and absorption. The paper by Van Dam et al (2018) will hopefully be of use.
2. To clarify the definition of ‘flow’ and its measurement.
3. To continue this conceptual clarification of the constructs in the discussion and conclusion section when interpreting the results and summarizing the findings.
4. To fix the consistency of the statistical reporting.
5. To consider the explanation of control group differences to explain the difference between the current findings and Fredborg et al.s, and include this if the authors agree.
6. To add a bit of extra discussion regarding the measurement of flow, and whether 43% of people reporting it is consistent with other literature (and if not, suggest possible theoretical or methodological reasons why).

·

Basic reporting

Please see my comments regarding extra literature, framing of the paper, and other comments in 'General Comments for the Author'.

Experimental design

Please see my comments on extra methodological details required in 'General Comments for the Author'.

Validity of the findings

Please see my suggestions for statistical analyses, as well as interpretation, discussion of the limitations of the study in 'General Comments for the Author'.

Additional comments

This paper reports the results of an online survey examining the relationships between absorption, flow, and mindfulness in ASMR. When compared to control participants, ASMR sensitive participants reported significantly higher rates of trait absorption but not flow or mindfulness (although ASMR participants were significantly more likely to have experienced flow). Additionally, the intensity of ASMR experience was positive correlated with trait absorption, suggesting that those with more intense ASMR triggers are also more likely to become highly absorbed more generally (this is a particularly nice finding that I think could be emphasised more in the discussion).

There is a lot to like about this paper: it is well written, suitably conducted, and reveals interesting and important results with regards to both differences between ASMR and non-ASMR participants but also individual differences within the ASMR sensitive population. I especially appreciate the distinction made between flow and ASMR and between mindfulness and ASMR because these concepts have been conflated in previous work (despite anecdotal reports that ASMR is different from both mindfulness and flow). This paper helps to make clear the connection between these experiences may be their underlying association with the trait of absorption.

I have no hesitation in recommending this paper for publication following a few revisions which I outline below. Overall, this is an excellent paper and a pleasure to review – thank you for contributing to our growing knowledge of this exciting experience!

1) Unnecessary removal of data points.

The rationale for reducing the number of data points in the analyses (which is such a waste of valuable data from willing participants!) is not clearly justified. The group size imbalance is not substantial (59% ASMR) and it is not clear how such an imbalance would adversely affect the validity of results or the conclusions drawn. The approach used would make sense if the number of participants was specified a priori and once that number of ASMR participants were recruited then data collection stopped. However, to remove valuable data after collection simply to ensure that the numbers of participants in each group are equal is unnecessary at best.

It also is not clear how participant groups were matched according to age and gender (i.e., the procedure for this – was it done randomly?) - so that would need to be clarified if the authors choose to keep the analyses with the subsample of the data. If age and gender are factors of concern then why not include them as covariates in an analysis rather than deleting data to ensure that the control group are age and gender matched? Or at least demonstrate in advance that age and gender are related in a meaningful way either to the experience of ASMR or to the dependent variables (through appropriate statistical analyses).

The authors should re-run their analyses using the full data set and report these in the text in addition to, or instead of, what is currently reported. If age and gender are of concern then I would suggest demonstrating this through statistical means and including them as covariates in an ANOVA. At the very least I would want to know that the results remain the same whether the full or reduced data set is used.

2) Citing previous literature.

Given that ASMR is such a small field it is important to fully present all the existing evidence related to the subject matter of the paper. I would therefore suggest including reference to other aspects of personality related to ASMR (e.g., link to neuroticism – Fredborg et al., 2017 – Frontiers and Poerio et al., 2018 PloS ONE supplementary materials). Another substantial oversight is a paper by Roberts et al., (2018, Psychology of Consciousness) which directly examined the relationship between ASMR, absorption and flow. These results should be included in both the introduction and discussion sections.

3) Framing of investigation and ASMR.

The objective of the study should be more clearly and accurately defined throughout the paper. In the introduction it is framed as an investigation into what underpins the ability to experience ASMR and its psychophysiological benefits. However, in the first line of the discussion it is framed as elucidating whether ASMR is associated with an atypical personality profile. I don’t think the paper does either of these things conclusively so perhaps the study is better framed as an investigation into personality differences between ASMR experiencers and non-experiencers?

I also question the use of the phrase ‘atypical’ in the sense that ASMR is referred to as being linked to an atypical personality profile – why is the term ‘atypical’ used? This could be a bit misleading for readers. I would suggest more neutral phrasing like: ‘Trait ASMR has been consistently associated with a number of personality traits including…’.

4) Consideration of Limitations.

There should be a discussion of the paper’s limitations for readers to appreciate. Three obvious limitations spring to mind but there may be others.

1) Causality – due to the correlational nature of the research it isn’t possible to conclude that the trait of absorption predisposes people to experience ASMR or that it underpins the experience of ASMR. The results could reflect third variable explanations as well as reverse causality (e.g., the ability to experience ASMR makes people more likely to get fully immersed in other experiences; both absorption and the ASMR trait are underlined by another variable such as openness to experience or dissociation)

2) Self-selection – recruited participants are likely to have known about the ASMR community in advance and likely to watch ASMR videos. Perhaps these people are more likely than non-video watchers to be high in trait absorption? In the methods section the ASMR questionnaire asks whether participants watch videos online but the results are not reported (or I could have missed them – but please report % if they haven’t been included).

Because you have data on whether participants watch ASMR videos or not I’d be really interested to see whether there is a difference between these two groups in terms of absorption – if there isn’t that might go some way towards allaying the self-selection concern.

3) Verification of ASMR status. A common criticism of ASMR research is the self-verification of ASMR status. This paper (line 110) notes that additional questions were asked of ASMR respondents but it isn’t clear what these questions were or how they were used to filter out people who may not truly experience ASMR (did you remove any participants after examining the responses to these questions?).

Please could these be described in the paper in full (as it will be useful for other researchers)? It may also be helpful to refer to methodological recommendations on recruitment for trait ASMR research (see Hostler et al., 2019, Peer J).

5) More minor points.

- Line 55: how does the atypical personality profile facilitate ASMR?
- In the introduction it would be helpful to describe common triggers earlier on for readers who are unfamiliar with ASMR.
- Either use ‘such as’ or e.g. but not both (this is in several places in the manuscript) e.g. line 70 and frequently in the methods section when describing example scale items.
- Consistency: the paper flips between US and UK English spelling.
- A parallel structure in the order of the results presented in text and Table 1 would be helpful for readers.
- In the results section it would make sense to have the descriptive data described first (i.e. response to the ASMR questionnaire).
- Please report exact p-values and 95%CI for effects.
- Calculation of ASMR intensity score needs to be clearer: does it take into account number of triggers and intensity or just intensity of selected triggers? For example if I have 2 triggers which are very intense (rated 6) would my average be 6 or would the average include 0 for all the other triggers that don’t induce my ASMR? Variety of triggers and intensity of triggers seem like two separate measures so perhaps keeping them separate would add more to our existing knowledge of individual differences in the ASMR experience? Could the authors please clarify their method of calculating intensity (perhaps intensity isn’t the best description for this measure if it also includes a measure of number of triggers?)?

---

## Round 0.2 · Minor Revisions

Thank you for your revised paper. The reviewers agree that this is an improvement on the previous version, but that there are a couple of points to address before we can make our final decision.

·

Basic reporting

No problem.

Experimental design

No problem.

Validity of the findings

No problem.

Additional comments

I am grateful to the authors for responding fully to my comments. The manuscript reads very well now, especially in light of the responses to the other expert reviewers.

·

Basic reporting

-

Experimental design

-

Validity of the findings

-

Additional comments

The authors have done an excellent job of responding well to the points raised in my first review – thank you. As an exception they have not fully addressed the point regarding how the study is able to able to shed light on the mechanisms of ASMR/the framing of the paper. This has much to do with how some (minor) sections of the paper have been written which set up incorrect expectations about the inferences that can be drawn from the methods used.

The introduction is set up as trying to understand mechanisms but the nature of the study (correlation) can’t answer these questions. I appreciate that the point about correlation as a limitation is now mentioned but I think there needs to be greater clarity about what the paper is able to achieve from the outset.

In the introduction the following statement is made:
“However, it remains to be established what underpins the ability to experience ASMR and its psychophysiological benefits. One line of evidence suggests that ASMR-experiencers show a specific personality profile which may facilitate ASMR.”

It still isn’t clear what it is about personality that facilitates ASMR. I suggest removing these sentences and replacing with something like ‘Previous research suggests that there are personality differences between people who experience ASMR and those who do not’. That way the reader isn't being led to believe that the paper will be able to answer questions about mechanisms.

It would also help to avoid confusion for readers if there was a distinction between state and trait ASMR. The phrase ‘ability to experience ASMR’ is used a lot through the manuscript but it isn’t clear whether this is meant in state or trait terms (or both?). It would help to make this clear in the paper in case readers thought that the paper sheds light on why some people experience ASMR and others don’t.

At the end of the discussion the following statement is made:
“In summary, our findings contribute to the existing literature documenting individual differences underpinning ASMR by showing that absorption proneness may be an important factor contributing to the ability to experience ASMR and its therapeutic benefits.”

This sounds good but it is slightly misleading. The word underpinning and general phrasing are suggestive of a mechanistic understanding of ASMR which is not possible from the methods used. It is also not possible to make any conclusions about the therapeutic benefits of ASMR as this was not assessed. It would help to reword so that the contribution is not overstated and suggestive of a mechanistic understanding of ASMR.

I say all of this because the paper is likely to be read by the general public/media as well as scientists due to the popularity of ASMR. It is therefore important not to overstate the potential of the methods for answering big mechanistic questions like ‘why do some people experience ASMR and others don’t?’ in case the results are taken out of context.

---

## Round 0.3 · accepted · Accept

Thank you for your revised version of the manuscript. I am delighted to inform you that we are happy to accept the paper for publication.